# *IbMYB308*, a Sweet Potato R2R3-MYB Gene, Improves Salt Stress Tolerance in Transgenic Tobacco

**DOI:** 10.3390/genes13081476

**Published:** 2022-08-18

**Authors:** Chong Wang, Lianjun Wang, Jian Lei, Shasha Chai, Xiaojie Jin, Yuyan Zou, Xiaoqiong Sun, Yuqin Mei, Xianliang Cheng, Xinsun Yang, Chunhai Jiao, Xiaohai Tian

**Affiliations:** 1College of Agriculture, Yangtze University, Jingzhou 434025, China; 2Institute of Food Crops, Hubei Academy of Agricultural Sciences, Wuhan 430072, China; 3College of Agriculture and Animal Husbandry, Qinghai University, Xining 810016, China; 4Key Laboratory for Quality Regulation of Tropical Horticultural Plants of Hainan Province, College of Horticulture, Hainan University, Haikou 570228, China

**Keywords:** *IbMYB308*, *Ipomoea batatas*, salt stress, overexpression, tobacco

## Abstract

The MYB (v-myb avian myeloblastosis viral oncogene homolog) transcription factor family plays an important role in plant growth, development, and response to biotic and abiotic stresses. However, the gene functions of MYB transcription factors in sweet potato (*Ipomoea batatas* (L.) Lam) have not been elucidated. In this study, an MYB transcription factor gene, *IbMYB308*, was identified and isolated from sweet potato. Multiple sequence alignment showed that IbMYB308 is a typical R2R3-MYB transcription factor. Further, quantitative real-time PCR (qRT-PCR) analysis revealed that *IbMYB308* was expressed in root, stem, and, especially, leaf tissues. Moreover, it showed that *IbMYB308* had a tissue-specific profile. The experiment also showed that the expression of *IbMYB308* was induced by different abiotic stresses (20% PEG-6000, 200 mM NaCl, and 20% H_2_O_2_). After a 200 mM NaCl treatment, the expression of several stress-related genes (*SOD*, *POD*, *APX*, and *P5CS*) was upregulation in transgenic plants, and the CAT activity, POD activity, proline content, and protein content in transgenic tobacco had increased, while MDA content had decreased. In conclusion, this study demonstrated that *IbMYB308* could improve salt stress tolerance in transgenic tobacco. These findings lay a foundation for future studies on the R2R3-MYB gene family of sweet potato and suggest that *IbMYB308* could potentially be used as an important positive factor in transgenic plant breeding to improve salt stress tolerance in sweet potato plants.

## 1. Introduction

Sweet potato (*Ipomoea batatas* (L.) Lam) is the seventh most important crop in the world and the fourth most important in China [1]. It is a primary source of starch, calories, proteins, anthocyanin, and minerals [1]. Soil salinity and drought are the most important abiotic stresses that limit crop growth, yield, and quality [2]. In China, sweet potato is mainly planted in barren and arid areas. Therefore, it is critical to improve sweet potato resistance to salt and drought. Experiments have shown that the overexpression of functional genes in sweet potato can significantly improve its tolerance to abiotic stresses [2,3,4,5,6].

Transcription factors (TFs) are key regulators that have crucial functions in response to various stresses [7]. MYB transcription factors are one of the largest protein families in plants and are characterized by highly conserved N-terminal MYB DNA-binding domain repeats that form three α-helices, the second and third of which are involved in the formation of a helix–turn–helix (HTH) fold [8,9]. A highly conserved MYB DNA-binding domain repeat encodes proteins with 51–53 amino acids (AAs) [9]. The MYB TFs are further divided into four groups—1R-MYB/MYB-related, R2R3-MYB, 3R-MYB (R1R2R3-MYB), and 4R-MYB—based on the position and number of MYB domain repeats. The R2R3-MYB subfamily is the most abundant among the four groups and has been an important target for improving tolerance to abiotic stresses in plants [10,11]. Members of this subfamily are divided into 23 subgroups based on their DNA binding and their retention of amino acid motifs in the C-terminal domain [11]. MYB genes in various plant species have been widely reported to improve tolerance to biotic and abiotic stresses, such as *Arabidopsis thaliana* [12,13,14], *Oryza sativa* [15,16,17], *Zea mays* [18,19,20], *Triticum aestivum* [21,22,23], *Brassica napus* [24,25], *Malus × domestica* [26,27], *Fragaria × ananassa* [28], etc. An R2R3-MYB gene, *AtMYB20*, enhances salt stress tolerance in *Arabidopsis thaliana* by downregulating the expression of PP2Cs [29]. *AtMYB2* enhances tolerance to drought mainly by activating abscisic acid (ABA)-mediated signaling pathways [30]. The overexpression of *AtMYB96* in *Arabidopsis*, through coordinated auxin and abscisic acid signal pathways, can increase drought tolerance. The overexpression of *GmMYB84* enhances soybean drought tolerance by increasing antioxidant enzyme activities and root elongation [31]. Salt stress is among the major environmental stresses that lead to plant growth restriction and yield reduction. Salt stress can lead to ionic stress, osmotic stress, and secondary stresses, especially oxidative stress [32]. *OsMYB91*, an R2R3-MYB gene of *Oryza sativa*, was confirmed to be involved in the tolerance to salt stress and plant growth by enhancing the capacity to scavenge active oxygen (ROS) [33]. An R2R3-MYB gene, *TaSIM*, enhanced salt tolerance when overexpressed in *Arabidopsis* [21]. In strawberry, the overexpression of *FvMYB24* upregulated the expression of several stress-related genes in response to salt stress, thus enhancing the tolerance of transgenic *Arabidopsis* [28]. Many R2R3-MYB genes have been reported to be involved in the salt stress response of plants, but the characterization of sweet potato *MYB* genes that participate in environmental stress responses has been inadequate compared with other plant species.

In this study, we identified and isolated an R2-R3 MYB family gene, *IbMYB308*, from sweet potato, and the expression pattern and function of the gene were analyzed. Moreover, we developed *IbMYB308* overexpression lines, and the contribution of *IbMYB308* in response to salt stress in transgenic tobacco was analyzed. This study will potentially contribute to the improvement of sweet potato tolerance to abiotic stresses and of the molecular breeding of crops.

## 2. Materials and Methods

### 2.1. Plant Materials

The drought-tolerant sweet potato material *cv.* Eshu11 was used to clone *IbMYB308* and to characterize its function. Wild-type *Nicotiana tabacum cv.* Wisconsin 38 (W38) was used for plant transformation. Tobacco was grown in a greenhouse (22 °C, 16/8 h day/night cycle). The sweet potato material *cv.* Eshu11 was also cultivated in a greenhouse (28 °C, 16/8 h light/dark period).

### 2.2. Cloning and Bioinformatics Analysis of IbMYB308 and Its Promoter

The total RNA of Eshu11 was extracted using TRIzol reagent (Tiangen, Wuhan, China) by following the manufacturer’s instructions. The genomic DNA of Eshu11 was extracted by the CTAB method, and the genomic DNA was stored at −20 °C. The quality and quantity of the RNA and DNA were visualized by 1% agarose gel electrophoresis, then a Nano-Drop ND-1000 spectrophotometer (Thermo Fisher Scientific, Wilmington, MA, USA) was used for quantification at optical densities of 260 and 280 nm, respectively. The first-strand cDNA synthesis was performed using *TransScript*^®^ All-in-One First-Strand cDNA Synthesis SuperMix for qPCR (One-Step gDNA Removal) (TransGen, Wuhan, China). Each 20 μL contained 4 μL 5 × *TransScript*^®^ Uni All-in-One SuperMix for qPCR, 1 μL gDNA Remover, 1 μg total RNA, and variable amounts of RNase-free water. Then, the process of PCR was as follows: 42 °C for 15 min, then 80 °C for 5 s. The full cDNA sequence of *IbMYB308* was obtained based on previous transcriptome sequencing. Specific primers *IbMYB308*-F/R (Appendix A) were designed based on *IbMYB308* transcriptome sequencing. The PCR reaction system contained 5 μL 10 × PAGE buffer (Mg^2+^), 8 μL dNTPs Mix, 1 μL primer *IbMYB308*-F/R, 0.5 μL *LA* Taq polymerase, 2 μL cDNA, and 32.5 μL ddH_2_O. The PCR procedure comprised an initial preheating step at 94 °C for 5 min, followed by 35 cycles of denaturation at 94 °C for 30 s, annealing at 55 °C for 30 s, and extension at 72 °C for 90 s, with a final extension at 72 °C for 10 min. The PCR products were separated via electrophoresis on a 1% agarose gel, and the target DNA fragments were recovered using an EasyPure^®^ Quick Gel Extraction Kit (TransGen, Wuhan, China) according to the manufacturer’s instructions. The resulting fragments were cloned into the pMD19-T vector (TakaRa, Wuhan, China) and sequenced by TIANYI Company (TIANYI, Wuhan, China).

The *IbMYB308* promoter sequence was cloned using the genomic walking method. *IbMYB308* promoter-specific primers *IbMYB308*pro-F/R (Appendix A) were designed according to the upstream 1431 bp sequence of *IbMYB308*. The PCR reaction system contained 5 μL 10 × PAGE buffer (Mg^2+^), 8 μL dNTPs, 1 μL primer *IbMYB308*pro-F/R, 0.5 μL *EasyTaq*^®^ polymerase, 2 μL genomic DNA, and 32.5 μL ddH_2_O. The PCR products were separated via electrophoresis, and the target DNA fragments were recovered using an *EasyPure*^®^ Quick Gel Extraction Kit (TransGen, Wuhan, China) according to the manufacturer’s instructions. All the resulting fragments were cloned into the pMD19-T vector (TaKaRa, Wuhan, China) and then transformed into competent *Escherichia coli* strain *DH5α* cells and sequenced by TIANYI Company (TIANYI, Wuhan, China).

*IbMYB308*-F/R primers were used to amplify the genomic DNA with the following procedure: initial preheating step at 94 °C for 5 min, followed by 35 cycles of denaturation at 94 °C for 30 s, annealing at 55 °C for 60 s, and extension at 72 °C for 90 s, with a final extension at 72 °C for 10 min. The PCR reaction system contained 5 μL 10 × PAGE buffer (Mg^2+^), 8 μL dNTPs, 1 μL primer *IbMYB308*-F/R, 0.5 μL EasyTaq^®^ polymerase, 2 μL genomic DNA, and 32.5 μL ddH_2_O. The PCR products were separated via electrophoresis on a 1% agarose gel, and the target DNA fragments were recovered using an EasyPure^®^ Quick Gel Extraction Kit (TransGen, Wuhan, China) according to the manufacturer’s instructions. The resulting fragments were cloned into the pMD19-T vector (TakaRa, Wuhan, China) and sequenced by TIANYI Company (TIANYI, Wuhan, China).

### 2.3. Bioinformatics Analysis

The *IbMYB308* sequence was determined using the BLAST-Protein program in the National Center for Biotechnology Information (NCBI) database (https://blast.ncbi.nlm.nih.gov/Blast.cgi?PROGRAM=tblastn&PAGE_TYPE=BlastSearch&LINK_LOC=blasthome accessed on 7 July 2022). DNAMAN software (version 8.0, Lynnon Biosoft, San Ramon, CA, USA) was used to compare the homologous sequence of *IbMYB308* in sweet potato and other species. Multiple sequence alignments were performed using MEGA X, and the phylogenetic tree was constructed using the neighbor-joining method with the 1000 bootstrap method. *Cis*-acting elements of the *IbMYB308* promoter sequence were analyzed using PlantCARE (http://bioinformatics.psb.ugent.be/webtools/plantcare/html/ accessed on 7 July 2022). The conserved domain was identified using SMART (http://smart.embl-heidelberg.de/smart/set_mode.cgi?NORMAL=1 accessed on 7 July 2022).

### 2.4. Expression Analysis of IbMYB308 in Sweet Potato and Transgenic Tobacco

The expression patterns of *IbMYB308* were analyzed in different tissues of sweet potato. The specific primers *IbMYB308*-F1/R1 (Appendix A) used for qRT-PCR were designed by Primer Premier 5. The expression profiles of *IbMYB308* were determined under different abiotic treatments. Four-week in vitro-grown Eshu11 plants were treated in half- Hoagland solution with 200 mM NaCl, 20% polyethylene glycol-6000 (PEG-6000), and 20% H_2_O_2_, respectively. There were three biological replicates for each experiment and three technical replicates for each sample. Sweet potato leaves were collected at 0, 1, 3, 6, 12, and 24 h after treatment and quickly frozen in liquid nitrogen. The timing of the stress treatments was determined according to Zhang’s experimental methods [5]. The first-strand cDNA synthesis was performed using TransScript^®^ All-in-One First-Strand cDNA Synthesis SuperMix for qPCR (One-Step gDNA Removal) (TransGen, Wuhan, China). According to the manufacturer’s instructions, TransStart^®^ Green qPCR SuperMix (50 × 20 μL reactions) includes 2 × TransStart^®^ Green qPCR SuperMix, Passive Reference Dye (50×), and nuclease-free water. Each 10 μL mixture contained 5 μL TransStart^®^ Green qPCR SuperMix (TransGen, Wuhan, China), 0.4 μL of each specific primer, 3.2 μL nuclease-free water, and 1 μL cDNA. The qPCR program comprised preheating at 94 °C for 2 min, followed by 45 cycles of denaturation at 94 °C for 5 s, and annealing at 58 °C for 30 s. The expression levels of *IbMYB308* were detected using qRT-PCR analysis conducted on the 7500 Real-Time PCR system (Applied Biosystems, Foster City, CA, USA). *IbActin* and *NtActin* genes were employed as the internal reference genes of sweet potato and tobacco, respectively. The specific primers are shown in Appendix A.

### 2.5. Construction of Overexpression Vectors

*IbMYB308*-F2/R2 primers (Appendix A) were used to amplify the open reading frame (ORF) of *IbMYB308*, and the sequence was inserted into the pCAMBIA1300-GFP vector, with the CaMV 35S promoter, between the *KpnI* and *BamHI* restriction sites to construct the overexpression plasmid of *IbMYB308* (Appendix A). The resulting pCAMBIA1300-*IbMYB308*-GFP overexpression plasmid was confirmed by sequencing and then transformed into tobacco (strain *GV3101* competent cells) to further explore the function of *IbMYB308*.

### 2.6. Generation of Transgenic Tobacco

Transgenic tobacco overexpression lines (OE lines) were obtained by transferring the pCAMBIA1300-*IbMYB308-GFP* with CaMV 35S promoter recombinant plasmid into *Agrobacterium tumefaciens* EHA105 using the freeze–thaw method. The tobacco was transformed by the leaf disc method [5]. Infected tobacco leaf discs were inoculated on MS medium containing 15 mg/L Hyg, 400 mg/L cephalexin, 1.0 mg/L 6-BA, and 0.1 mg/L NAA in the dark at 27 ± 1 °C for 30 d, and then the regenerated shoots were transferred to 1/2 MS medium with 25 mg/L Hyg and 100 mg/L Carb for the formation of whole plants. Primers 35S-F/1300-R (Appendix A) were used to detect *IbMYB308* overexpression in the tobacco plants. Genomic DNA was extracted from the leaves of transgenic and wild-type (WT) plants and amplified under the following conditions: preheating at 95 °C for 5 min, followed by 35 cycles of denaturation at 95 °C for 30 s, annealing at 62 °C for 30 s, extension at 72 °C for 1 min, and finally extension at 72 °C for 10 min. The PCR products were detected by 1% agarose gel electrophoresis to confirm the insertion of *IbMYB308* into the transgenic plants. Further, qRT-PCR was used to detect the expression of *IbMYB308* in transgenic plants, with the *NtActin* gene used as the internal reference gene of *Nicotiana tabacum* [34]. Transgenic lines were selected for phenotypic investigation and WT tobacco was used as the control.

In vitro identification of the salt tolerance of the transgenic tobacco plants was based on the method of Zhang [5]. The transgenic and WT tobacco plants were grown in normal conditions and in MS medium with 200 mM NaCl. The culture conditions were 27 ± 1 °C, 13 h per day. After being cultured with a NaCl treatment for four weeks, the growth status of the transgenic plants was observed, and the content of proline and protein and the activity of CAT, MDA, and POD were determined. The expression levels of the abiotic stress-responsive genes were determined in the transgenic tobacco plants both under normal conditions and with a 200 mM NaCl treatment. The abiotic stress-responsive genes included *SOD*, *POD*, *APX*, and the proline synthesis-related gene, *P5CS*. The genes’ specific primers were designed by Primer Premier 5. The primer sequences are shown in Appendix A.

### 2.7. Data Analysis

The experiments were set in three biological replicates for each experiment and three technical replicates for each sample. IBM SPSS Statistics 26 software was used for statistical analysis; the data were analyzed using one-way ANOVA, two-way ANOVA, or Student’s two-tailed *t*-test, and the results are presented as the mean ± standard deviation, with the significance level at *p* < 0.05 or *p* < 0.01.

## 3. Results

### 3.1. Isolation and Characterization of IbMYB308

One differential expressed sequence selected from the transcriptome sequence data of drought-tolerant sweet potato Eshu11 was used to clone the *IbMYB308* gene by the amplification of cDNA. The 844 bp full-length cDNA of *IbMYB308* contains a 768 bp open reading frame (ORF) that encodes 255 amino acids. The predicted molecular weight (MW) of the resulting protein was 28.95 KDa, and the deduced isoelectric point (PI) and instability index were 6.99 and 56.80, respectively. The genomic sequence of *IbMYB308* is 1101 bp in length and contains two exons and one intron (Figure 1a). The subcellular localization prediction analysis of the protein encoded by the *IbMYB308* gene in sweet potato showed that the protein was mainly distributed in the nucleus.

Protein sequence alignment analysis showed that the IbMYB308 protein sequence was highly homologous to the MYB308 or MYB308-like protein of *Olea europaea*, *Pistacia vera*, *Gossypium hirsutum,* and *Juglans regia*, indicating that MYB308 was conserved in plants. The IbMYB308 protein contained two typical SANT (R2, R3) domains at the N-terminus, but there was sequence diversity at the C-terminus, indicating that IbMYB308 is an R2R3-type transcriptions factor. To identify the evolutionary relationship of IbMYB308, a phylogenetic tree containing 34 AtMYBs from *Arabidopsis thaliana*, seven OsMYBs from *Oryza sativa*, one ItMYB from *Ipomoea triloba*, one InMYB from *Ipomoea nil*, and one NtMYB from *Nicotiana tabacum* was constructed with MEGA X (Figure 1b). IbMYB308 belonged to the I subfamily. Furthermore, the promoter of *IbMYB308* was cloned and analyzed. The full length of the *IbMYB308* promoter was 1431 bp, and it contained one element involved in light responsiveness (G-box), one light-responsive element (3-AF1 binding site), one element involved in abscisic acid responsiveness (ABRE), one gibberellin-responsive element (GARE-motif), one element involved in salicylic acid responsiveness (TCA-element), three Myb-binding sites, and one element related to meristem expression (CAT-box), indicating that *IbMYB308* may be related to plant stress response (Appendix A).

The motifs of *IbMYB308* and other *MYB308* or *MYB308-like* species were identified by MEME software. The six motifs comprised between 11 and 50 amino acids. Sequence analysis showed that motif one and motif two each contained the complete structure of the Myb-like DNA binding domain. Other motifs were conserved domains of MYB transcription factors. It was revealed that IbMYB308 was a typical R2R32-MYB transcription factor (Figure 2).

### 3.2. Expression Analysis of the IbMYB308 in Sweet Potato

The expression patterns of *IbMYB308* were analyzed by qRT-PCR, which showed that *IbMYB308* was expressed in the root, stem, and leaf tissues of sweet potato, but the expression levels differed among tissues. The expression level of *IbMYB308* in the leaf were significantly higher than in the root and stem (Figure 3).

### 3.3. Expression Profiles of IbMYB308 under Abiotic Stress

To explore the effect of stress on *IbMYB308* expression, qRT-PCR was used to determine the expression levels of *IbMYB308* in sweet potato under different abiotic stresses. There were three stress treatments—200 mM NaCl, 20% PEG-6000, and 20% H_2_O_2_—for analysing the expression profiles of *IbMYB308*. For the 200 mM NaCl treatment, the expression increased, reaching a peak at 3 h, and then gradually decreased, though remained higher than expression at 0 h (Figure 4b). The expression of *IbMYB308* was significantly higher at 3 h and 6 h than at 0 h. Under 20% H_2_O_2_ treatment, the expression levels decreased at 1 h, then increased at 3 h, 6 h, and 12 h, reaching its peak at 12 h, and then decreased, with expression lower at 24 h than 0 h. The expression of *IbMYB308* was significantly higher at 6 h and 12 h than 0 h (Figure 4c). Under PEG-6000 treatment, expression of *IbMYB308* was lower at 1 h, 3 h, 6 h, 12 h, and 24 h than 0 h (Figure 4a).

### 3.4. Overexpression of IbMYB308 Improves Tolerance to Salt Stress in Transgenic Plants

To further analyze the function of *IbMYB308,* the overexpression vector pCAMBIA1300-*IbMYB308*-*GFP* was constructed and transferred into the WT tobacco plants. DNA extracted from the leaves of transgenic lines was used as templates; the *IbMYB308* overexpression lines were identified by PCR, and the WT plants, *IbMYB308* plasmid, and ddH_2_O were used as controls (Appendix A). Three lines with high expression levels (OE-3, OE-6, and OE-7; Figure 5) were selected for further functional analysis. The WT plants were used as a control.

To verify the response of overexpressing *IbMYB308* tobacco to salt stress, the three transgenic lines (OE-3, OE-6, and OE-7) and the WT plants were grown both in MS medium with 200 mM NaCl and in stress-free conditions for four weeks. The WT and transgenic lines showed no significant difference in growth status under stress-free conditions, but the growth status of the transgenic lines was better than that of the WT under salt stress (Figure 6a). Malondialdehyde (MDA) content in plants reflected the degree of plant damage [35,36]. CAT activity, POD activity, and proline content in plants reflected the antioxidant capacity of plants [37]. Under salt treatment, the content of MDA in the transgenic lines and the WT was elevated compared with that under normal treatment, and the content of MDA in the three transgenic lines (47.710 ± 1.020, 44.992 ± 1.668, and 42.390 ± 1.102 μmol/g·Fw, respectively) was significantly lower than that in the WT (55.6827 ± 2.067 μmol/g·Fw) (Figure 6d). The CAT activity in the three transgenic lines was higher than in the WT under normal treatment. Moreover, under NaCl treatment the CAT activity in the three transgenic lines (2.133 ± 0.015, 2.217 ± 0.015, and 2.380 ± 0.069 U/g·Fw, respectively) was significantly higher than in the WT (Figure 6b). The POD activity in the transgenic lines was markedly higher than in the WT under NaCl treatment (Figure 6c). Among the three transgenic lines, the POD activity of OE-7 (203.56667 ± 1.925 10^3^ U/g·Fw) was twice that of the other lines. The proline content of transgenic lines and the WT were approximately at the same level under normal treatment (Figure 6e). Transgenic lines OE-6 (61.110 ± 1.593 μg/g·Fw) and OE-7 (70.111 ± 1.736 μg/g·Fw) had significantly higher proline content than the WT under NaCl treatment. Under normal treatment, the transgenic and WT tobacco had similar proline content. However, the protein content in the transgenic lines was higher than that in the WT under salt stress (Figure 6f). These results showed that transgenic lines have better resistance to salt stress and damage than WT plants.

Under the stress of a 200 mM NaCl treatment, the expression levels of the abiotic stress-responsive genes, *SOD*, *POD*, and *APX*, and the proline synthesis-related gene, P5CS, were upregulated in OE-lines compared with those in WT plants (Figure 7).

## 4. Discussion

Transcription factors are key in regulating gene expression. Abiotic stress is one of the most important factors that limit plant growth and productivity worldwide. Plants have evolved various mechanisms to adapt to environmental changes at different levels, including pressure signal sensing and transduction, the activation of specific transcription factors, and the expression of related genes [36]. Therefore, the breeding of plant cultivars resistant to abiotic stresses has become a major goal for agricultural development. R2R3-type MYB family members have been identified in *Arabidopsis* [11], *Oryza sativa* [38], *Ananas comosus* [39], *Nicotiana tabacum* [40], *Dimocarpus longan* [41], and *Camellia sinensis* [42]. However, sweet potato R2R3-MYB genes associated with salt tolerance have rarely been studied. In this study, an R2R3-MYB transcription factor, *IbMYB308*, was isolated from sweet potato, which was strongly induced by NaCl (Figure 4b). According to the phylogenetic analysis, IbMYB308 was clustered into group I and classified with MYBs from *Ipomoea triloba* ItMYB308 (Figure 1c). Multiple sequence analysis revealed that the IbMYB308 protein shared two Myb-like DNA binding domains with the MYB308 or MYB308-like protein sequence of other plants. It was revealed that IbMYB308 was a typical R2R3-MYB transcription factor (Figure 2). Prediction of subcellular localization suggests that IbMYB308 is located in the nucleus; it is speculated that the transcription factor may be involved in the transcription levels of other genes.

Gene function can be reflected, to some extent, in the expression patterns of the gene [43]. *NsMYB1* had a higher transcription level in fruit and a lower expression level in root and leaf tissues, but it was hardly expressed in stem tissue [44]. *AtMYB74* was expressed in the root, stem, rosette leaf, cauline leaf, flower, and silique tissues of *Arabidopsis*. The expression levels were highest in flower, rosette leaf, and cauline leaf tissues [14]. *OsMYBc* had the highest expression in the leaf blade and lower expressions in leaf sheath, basal stem, and, especially, root tissues [45]. In *Gerbera hybrida*, the expression locations of *GhMYB1a* included bract, old leaf, young leaf, and stamen tissues, but the expression levels in young root, old root, scape, and pappus tissues were lowest [46]. In this study, the expression levels of *IbMYB308* were analyzed in root, stem, and leaf tissues. The expression levels of *IbMYB308* were significantly higher in leaf than root and stem tissues (Figure 3). The expression patterns of *IbMYB308* were not the same as *AtMYB71*, *OsMYBc*, and *GhMYB1a*, but all four genes showed higher expression levels in the leaf. The expression patterns of the *MYB308* genes varied from species to species. *PgMYB308-like* had a higher expression level in the root [47]; *PlMYB308* was expressed in petal, sepal, pistil, leaf, stamen, and stem tissues, with the expression levels in petals higher than in other tissues [48]. In sweet potato, the expression patterns of *IbMYB1* and *IbMYB116* were different. *IbMYB1* was involved in the regulation of anthocyanin biosynthesis in the leaves and storage roots, and *IbMYB116* was associated with drought resistance in sweet potato [49,50]. In the current study, *IbMYB308*, *IbMYB116*, and *IbMAM1.1* [51] had very similar expression patterns, and the expression levels were higher in leaf than root and stem tissues. High expression levels of *IbMYB308* were detected in leaf tissue. This gene may be related to the protection mechanism of sweet potato stress resistance. Moreover, it may be involved in the relevant regulatory network to enhance the stress resistance of leaf organs.

Promoters are gene switches, located upstream of gene coding regions, and contain specific *cis*-acting elements that play a crucial role in gene transcription and expression [52,53]. The main active position of *cis*-acting elements with a biological function is 50 bp upstream of the transcription start sites (TSS), and most transcription factor binding sites (TFBS) are in the region of −1000 bp to +200 bp relative to the TSS [54]. In this study, the 1431 bp upstream sequence of the *IbMYB308* promoter was cloned (Figure 1d). There were many TATA-boxes on the promoter of *IbMYB308*, which is a core promoter element located 25–35 bp upstream of the TSS [55]. A CAAT-box, which is a ubiquitous *cis*-acting element in promoters and located 75 bp upstream of the TSS, was present in the *IbMYB308* promoter [56]. There were other *cis*-acting elements present: G-box, ARE, ABRE, AT~TATA-box, MYB, CAT-box, and TCA-element (Figure 1d, Appendix A). These *cis*-acting elements were associated with abiotic stress and hormonal response. The *IbMYB308* promoter also contained Myb-binding sites. This too indicated that *IbMYB308* may be involved in abiotic stress and hormonal response, and in the binding of target genes. In wheat, *TaMYB344* and *TaMYB67* were induced by drought and salt stress [22]. The expression patterns of the *MYB* gene family in *Nicotiana tabacum* were different: *NtMYB38* and *NtMYB46* were induced by drought and salt stress, whereas *NtMYB36*, *NtMYB45*, and *NtMYB110* were induced by cold stress [40]. In pigeon pea, *CcMYB5* and *CcMYB14b* were strongly induced by drought, *CcCHS3* was induced under GA3 treatment, *CcCAD6* was induced by drought and salt stress, and, conversely, *CcCCR12* and *CcLAC1* were only induced by drought and salt stress, respectively [57]. The expression levels of *MYB* genes were different under different abiotic stresses (such as drought, cold, and salt stress), and the expression level also differed with the length of processing time [58]. *HpMYB48*, *HpMYB70*, and *HpMYB102* were upregulated by NaCl and drought at different points. For example, the expression levels of *HpMYB70* were highest after 3 h under NaCl treatment [59]. *IbMYB308* was induced by abiotic stress, NaCl, H_2_O_2_, and drought (Figure 4). Under the treatment of 200 mM NaCl, expression levels were upregulated and reached their peak after 3 h of treatment (Figure 4b). The expression levels of *IbMYB308* were induced by 20% H_2_O_2_, and the expression levels were upregulated at 3 h, 6 h, and 12 h. Compared with the expression levels at 0 h, the expression levels at 1 h and 24 h were downregulated (Figure 4c). However, the expression levels of *IbMYB308* were downregulated under 20% PEG-6000 treatment (Figure 4a). Moreover, the expression patterns of genes were different under the same treatments [59]. For example, *GhCBS4*, *GhCBS15*, and *GhCBS45* downregulated under PEG treatments. The expression levels of *GhCBS32* were upregulated under NaCl treatment and reached their highest levels at 4 h, then decreased [60]. In *Populus*, *PtMYBR133* and *PtMYBR056* were downregulated under drought stress; however, the expression levels of *PtMYBR133* were upregulated in the cold compared with the control [61]. These results suggested that *IbMYB308* have different expression patterns under different abiotic stresses and also provided a deeper understanding of *MYB* gene expression patterns in plants.

Under salt or drought stress, plants often produce a large amount of reactive oxygen species (ROS), such as superoxide anion (O^2−^) and hydrogen peroxide (H_2_O_2_) [62,63]. SOD and POD scavenging systems can detoxify ROS to reduce oxidation damage in plant cells and enhance the resistance to stresses [64,65]. Proline can regulate the pH of plant cytoplasm and protect the integrity of cell membranes. Proline also has the function of scavenging reactive oxygen species (ROS) [66]. MDA content can lead to cell membrane damage and reduced salt and drought tolerance in plants [36,67]. The activity and content of plant physiological indicators (CAT activity, PDO activity, MDA content, and proline content) can reflect the resistance to adversity and stresses [68]. In this study, *IbMYB308* overexpression tobacco plants were obtained to analyze the function of *IbMYB308*. Three transgenic lines (OE-3, OE-6, and OE-7) were selected for functional analysis (Figure 5). It was found that the transgenic and WT plants had the same growth status under normal treatment, but the growth of transgenic lines was better than that of the WT plant under NaCl treatment (Figure 6a). In this study, under salt stress conditions, transgenic tobacco plants with overexpression of *IbMYB308* had upregulation in several abiotic stress-responsive genes (*SOD*, *POD*, and *APX*), and a proline synthesis-related gene, *P5CS* (Figure 7). Under salt stress, both transgenic and WT lines were impaired, but there were huge differences in CAT activity, POD activity, MDA content, and proline content. CAT activity, POD activity, and proline content were higher in transgenic lines than in the WT, and MDA content was lower in transgenic lines than in the WT (Figure 6b–e). Protein plays an important role in plant response to adversity [69]. Transgenic lines have higher protein content than the WT plants under NaCl treatment (Figure 6f). Furthermore, it was verified that the *IbMYB308* overexpression in tobacco improved the salt stress tolerance compared with the WT. A simple hypothetical model of the regulatory mechanism of *IbMYB308* involved in the response to abiotic stress is shown in Figure 8. The function of *IbMYB308* was similar to other R2R3-MYB genes. *GhMYB73* transgenic *Arabidopsis* was more tolerant to salt stress [70]. Overexpression of *ThMYB8* improved the salt stress tolerance of transgenic *Arabidopsis* [71]. Heterologous expression of *CsMYB30* enhanced the tolerance of transgenic *Arabidopsis thaliana* to salt and drought stress [72].

In general, these data revealed that overexpression of *IbMYB308* enhanced the salt tolerance of transgenic tobacco.

## 5. Conclusions

R2R3-MYB transcription factors are one of the most important classes of transcriptional regulators in plants. In this study, an R2R3-MYB transcription factor gene from sweet potato, *IbMYB308*, was isolated from Eshu11. The expression patterns and functional characteristics of *IbMYB308* were investigated, and the expression levels of *IbMYB308* were induced with NaCl, H_2_O_2_, and PEG-6000 treatments. The overexpression of *IbMYB308* in tobacco improved the tolerance of transgenic tobacco plants to salt stress. These findings suggest that *IbMYB308* plays a role in salt stress responses. Moreover, this study is of potential value as a resource for salt-tolerant sweet potato breeding.

## Figures and Tables

**Figure 1 genes-13-01476-f001:**
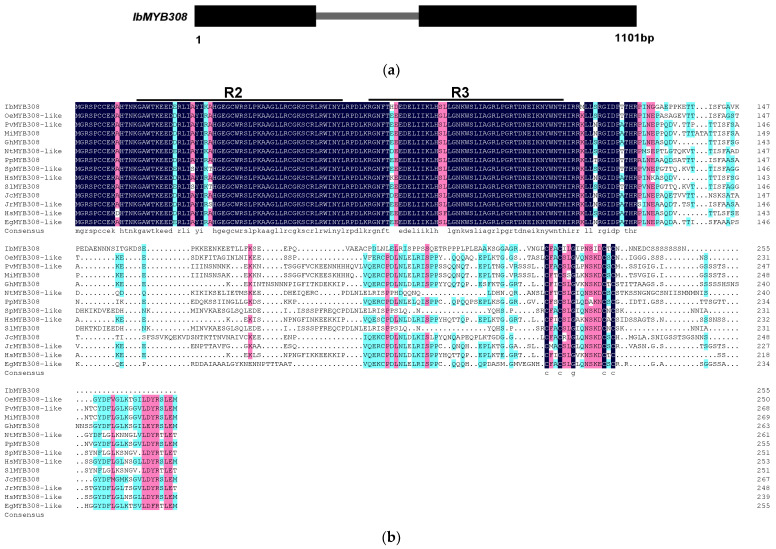
Characterization of *IbMYB308*. (**a**) Genomic structures of *IbMYB308*. Boxes indicate exons, and lines indicate introns. (**b**) Multiple protein sequence of IbMYB308 with R2R3-MYB DNA binding domains in *Olea europaea* (accession number CAA2964916.1), *Pistacia vera* (XP_031265728.1), *Mangifera indica* (XP_044488236.1), *Gossypium hirsutum* (XP_016708004.1), *Nicotiana tabacum* (NP_001311732.1), *Prunus persica* (XP_007200603.1), *Solanum pennellii* (XP_015066308.1), *Jatropha curcas* (XP_012081759.1), *Juglans regia* (XP_018823839.1), *Hibiscus syriacus* (XP_039065864.1), and *Eucalyptus grandis* (XP_039166990.1). The identical and similar amino acids are shaded in black, pink, and light blue, and conserved domains are marked by black lines. (**c**) Phylogenetic analysis of IbMYB308. MEGA X software with the neighbor-joining (NJ) method (1000 bootstrap repeats) was used to construct the phylogenetic tree. IbMYB308 is marked with a red box. (**d**) The *cis*-acting elements of the *IbMYB308* promoter.

**Figure 2 genes-13-01476-f002:**
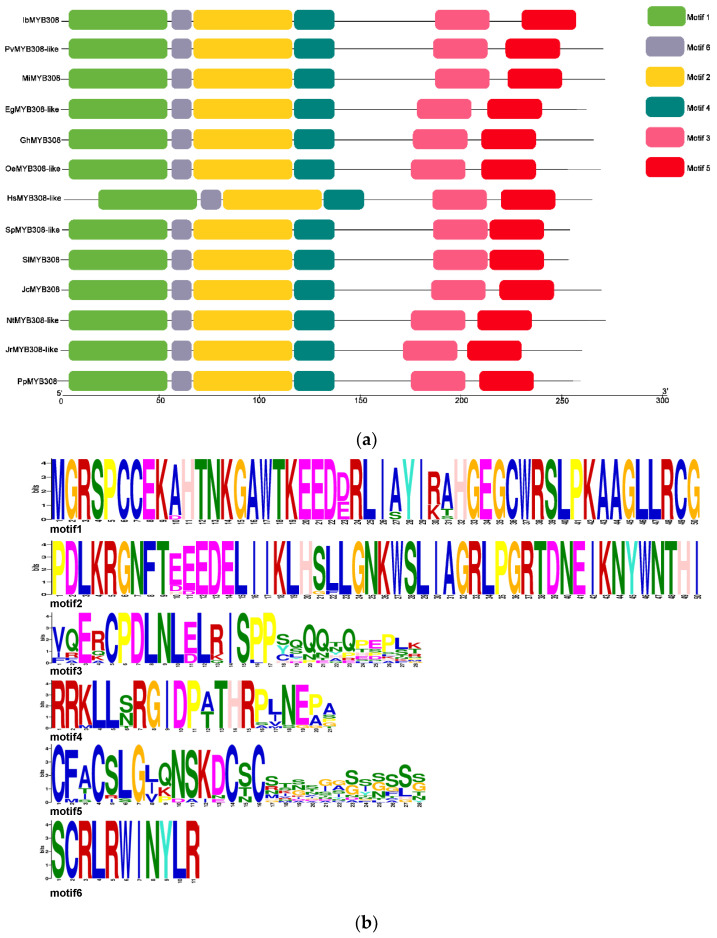
Motifs in IbMYB308 and MYB308 or MYB308-like species. (**a**) Motif location. (**b**) The width and amino acid sequence of conservative motifs.

**Figure 3 genes-13-01476-f003:**
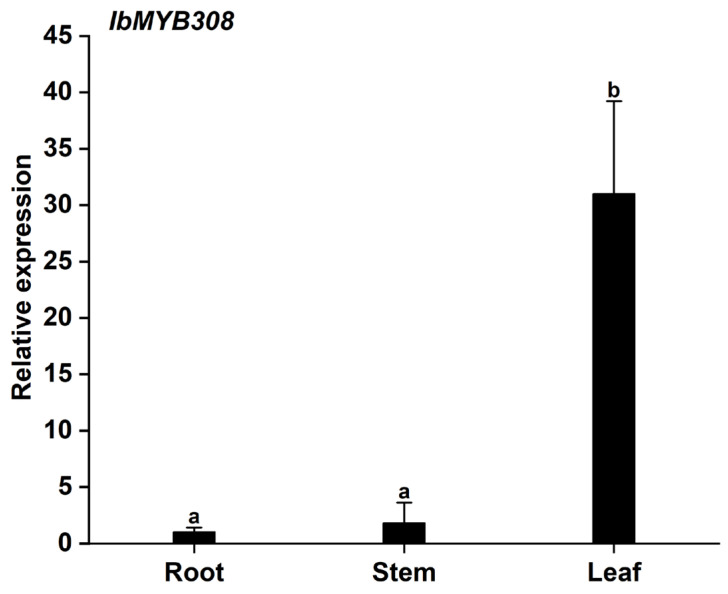
Expression of *IbMYB308* in different tissues of sweet potato by qRT-PCR. Data represent the mean of three biological replicates ± SD (*n = 3*). Error lines indicate standard deviations. Different lowercase letters (a, b) on the bars indicate significant differences at *p* < 0.01.

**Figure 4 genes-13-01476-f004:**
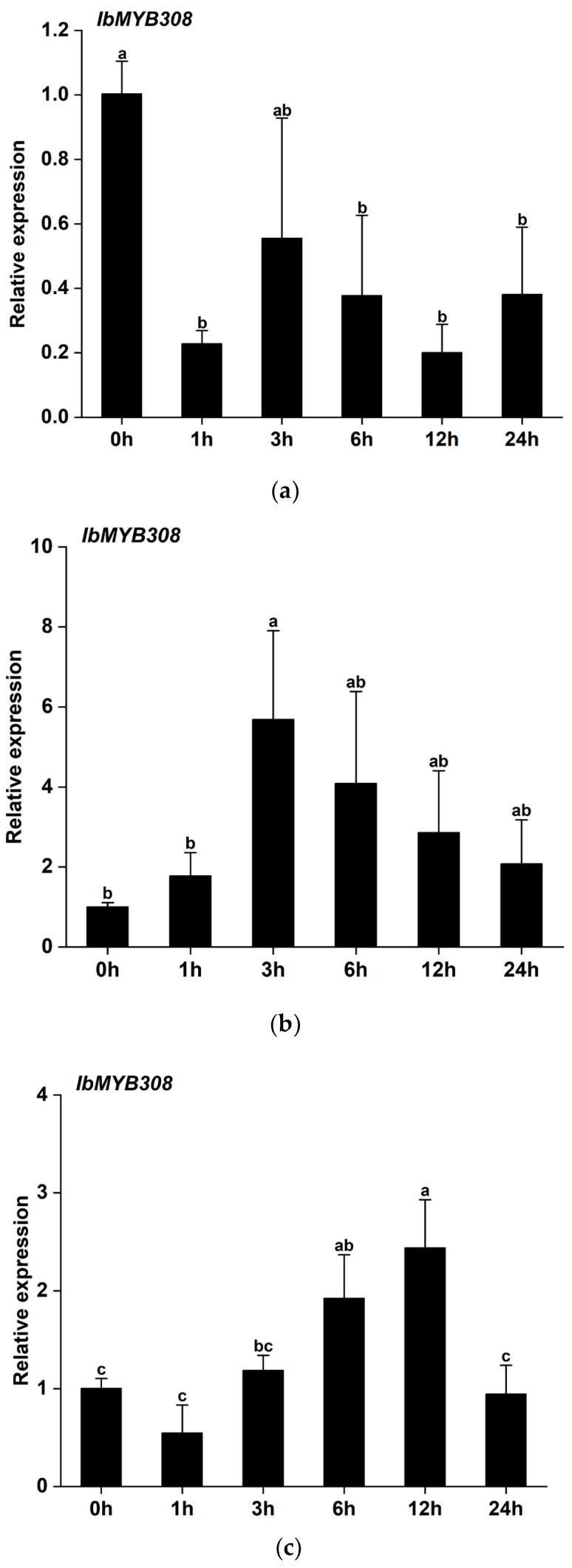
Expression profiles of *IbMYB308* in response to abiotic stress treatments. Sweet potato under (**a**) 20% PEG-6000, (**b**) 200 mM NaCl, and (**c**) 20% H_2_O_2_. *IbActin* was used as an internal reference gene. Data are presented as the means of three biological replicates ± SD (*n = 3*). Error lines indicate standard deviations. Different lowercase letters (a–c, ab, bc) on the bars indicate significant differences at *p* < 0.01.

**Figure 5 genes-13-01476-f005:**
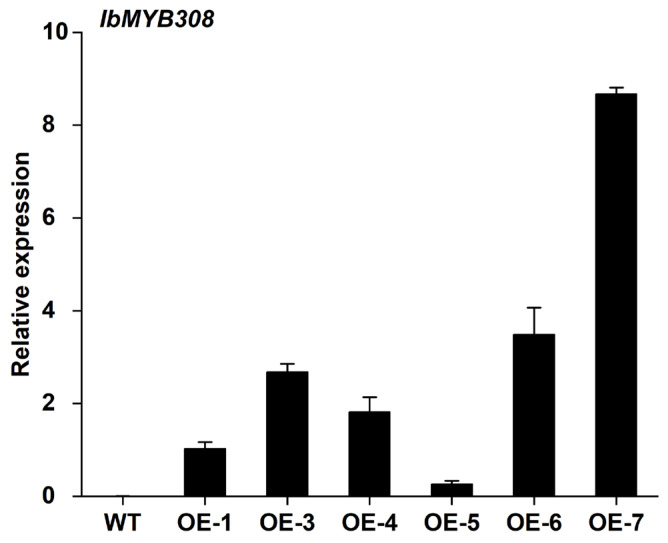
Identification by qRT-PCR of *IbMYB308* transcript in the six overexpression tobacco lines and the WT. *NtActin* was used as the internal reference gene. Bars represent SE from the mean of three technical replicates and three biological replicates. Data are presented as the means of three biological replicates ± SD (*n = 3*).

**Figure 6 genes-13-01476-f006:**
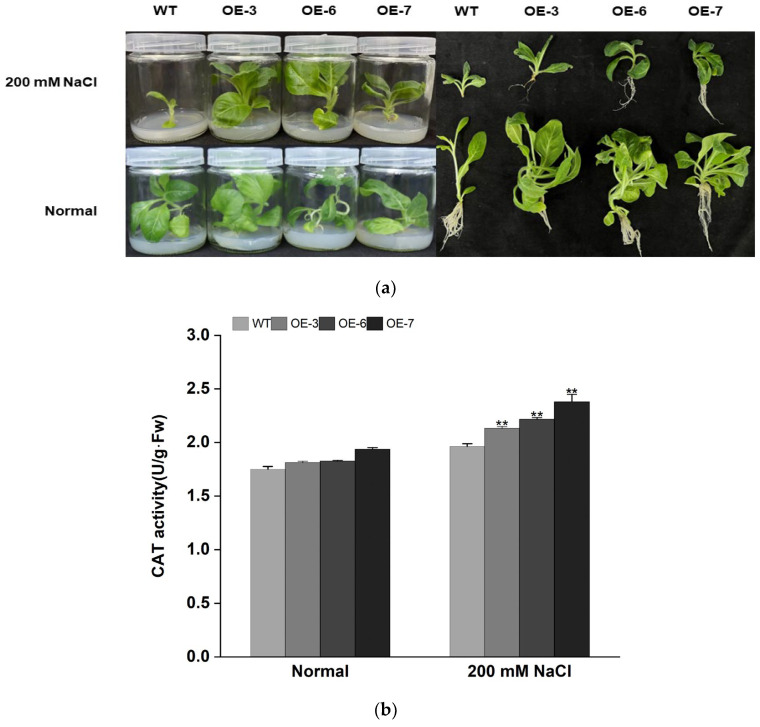
*IbMYB308* improves salt tolerance in transgenic tobacco plants. (**a**) Performance of *IbMYB308* transgenic tobacco and WT cultured for four weeks on MS medium without stress or with 200 mM NaCl. (**b**) CAT activity. (**c**) POD activity. (**d**) MDA content. (**e**) Proline content. (**f**) Protein contents of OE-lines and the WT under normal and 200 mM NaCl treatments. Data are presented as the means of three biological replicates ± SD (*n = 3*). ** indicates significant differences at *p* < 0.01.

**Figure 7 genes-13-01476-f007:**
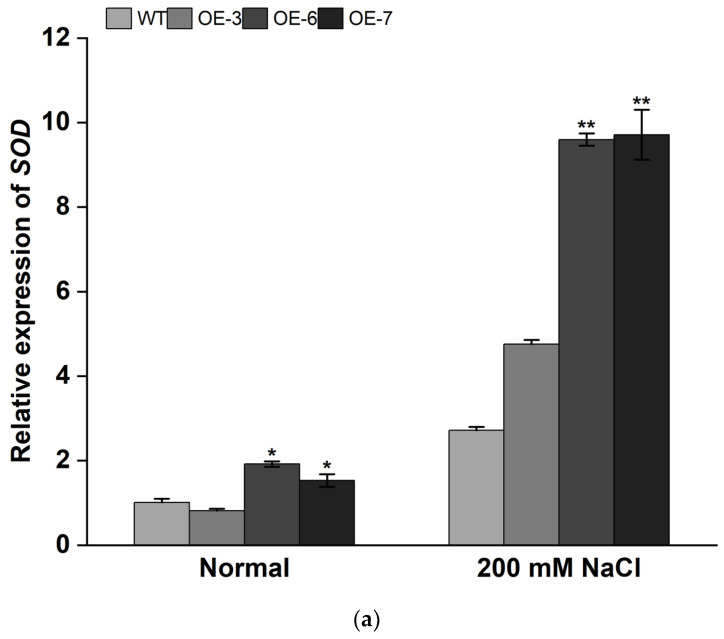
Relative expression of abiotic stress-responsive genes in the transgenic and WT tobacco plants. The abiotic stress-responsive genes of (**a**) *SOD*, (**b**) *POD*, (**c**) *APX*, and the proline synthesis-related gene, (**d**) *P5CS*. *NtActin* was used as the internal reference gene. Data are presented at the means of three biological replicates ± SD (*n = 3*). * and ** indicate significant differences at *p* < 0.05 and *p* < 0.01, respectively.

**Figure 8 genes-13-01476-f008:**
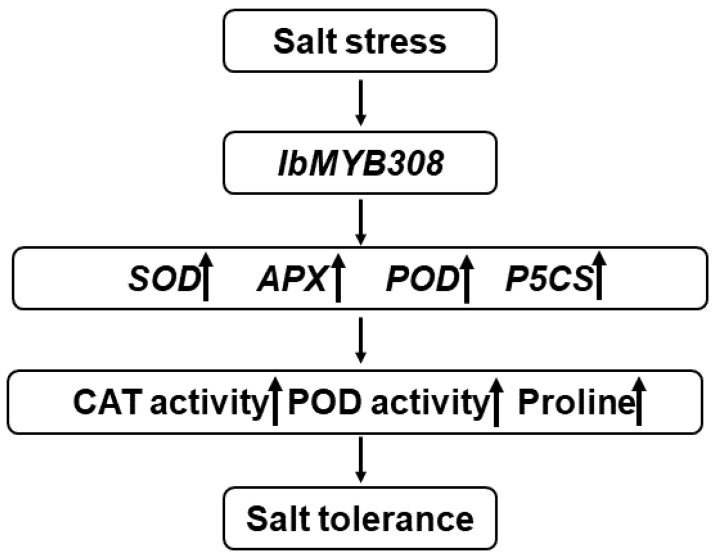
Schematic model of the *IbMYB308* regulatory mechanism of the abiotic stress response.

## Data Availability

Not applicable.

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
