# Peer review of "IbMYB308*, a Sweet Potato R2R3-MYB Gene, Improves Salt Stress Tolerance in Transgenic Tobacco"

_genes, 2022, doi:10.3390/genes13081476_

Round 1

Reviewer 1 Report

In the current manuscript, Wang et. al. have tried to explore the role of sweet potato MYB308 transcription factors in stress tolerance mechanism through heterologous expression of IbMYB308 in tobacco. 

The following points need to be addressed to further improve this manuscript.

 1. Title of the paper needs to be restructured to justify the content of the paper. 

2. Authors are suggested to include figures of transgenic tobacco lines they have developed. 

3. Check Table S1 and Line no. 179, primer listed ‘NtActin’ for its accuracy. 

4. Figure 1a. Box and lines are not clearly visible so suggested replacing this image with a better version. 

5. Figure 1b, is there any specific reason for an exclusive selection of MYB protein sequence from 13 different species only to show sequence alignment in figure 1b? If so, discuss it critically in the discussion section. 

6. Figure 1c, Why specifically, At, Os, and In MYB has been selected only for phylogenetic analysis of IbMYB 308 in Figure 1 c? Why has not NtMYB 308 been taken into consideration during phylogenetic analysis? 

7. Line 261-262, Restructure the statement for better understanding. 

8. Section 3.3, Which tissue of sweet potato was used to analyze expression under abiotic stress conditions? Moreover, the authors are suggested to include figures of sweet potato tissue undergoing such stress treatment. 

9. Authors are suggested to include pictures to show the result mentioned in line no. 299-303. 

10. Line No. 319, Check for the typo error. 

11. Section 3.4, Line 320-322, Which tissue of transgenic tobacco was considered for stress treatment? 

12. Line 355-356, Figure 1c, What is the basis and significance of classifying Cluster I, and how it is different from Cluster II? Critical discussion is required on it. 

13. In the discussion section, the authors have mentioned as other MYBs e.g., IBMYB116 have role in stress tolerance so whether authors have checked the expression of other MYBs to verify that the stress tolerance they have observed is exclusively due to MYB308 and not the cumulative effect of over/down expression of other MYB variants. 

14. Line no. 382-383, check for typo error. 

15. Line no. 370-374, What is the possible reason for such expression biases of MYB308 in different tissue? Authors are suggested to critically discuss their results in the discussion section rather than repeating the statement already mentioned in the result section. 

16. Line no. 458-461, MYB genes have been already established for their role in stress mechanism so the such broad statement should be avoided.

Author Response

Author‘s notes to reviewer are shown in attachment “Response to Reviewer 1 Comments”

Reviewer 2 Report

The study is well-conducted and provided important results that might use to improve salt tolerance in plants. However, some minor revisions are suggested as shown below;

 - The manuscript has few mistakes in English grammar and sentences structure, so I recommend the authors to used English proofreading service.

 - The abstract should highlight the most important results of the parameters and characteristics assayed.

 - In Material and Methods:- indicate how many replicates assayed in each analysis/parameter. Number of sample or biological and technical replicates should be mentioned for each parameter in the methods!

 - Results are explained in detail. 

- The discussion should be interpreted with the results as well as discussed in relation to the present literature.

Author Response

Author‘s notes to reviewer are shown in attachment “Response to Reviewer 2 Comments”

Reviewer 3 Report

To,

The Editor,

Gene, MDPI,

Manuscript ID: genes-1863094

 Subject: Submission of comments of the manuscript in “Life"

 Dear Editor Gene, MDPI,

 Thank you very much for the invitation to consider a potential reviewer for the manuscript (ID: genes-1863094). My comments responses are furnished below as per each reviewer’s comments. 

In the reviewed manuscript, authors were identified 21 an R2R3-MYB transcription factor gene family and isolated IbMYB308 from sweet potato. qRT-PCR analysis revealed IbMYB308 was differentially expressed in root, and stem, especially in leaf. Further, IbMYB308 was induced by different abiotic stresses including 20% PEG-6000, 24 200mM NaCl, and 20% H2O2. Furthermore, the heterologous expression of IbMYB308 in tobacco enhanced the tolerance to salt stress. Therefore, these results would help to understand the role of the R2R3-MYB gene and the underlying molecular mechanism of stress resistance in sweet potato. In general, the manuscript represents very big piece of research in a logical presentation. Therefore, it might be conditionally accepted with subject to major revision. However, this major revision means only that there is no necessity to repeat or extend the experiments and their analysis. Instead, authors have to improve their manuscript with many non-clear meaning, inaccuracy and inconsistencies, and the authors need to address the following issues before it can be accepted for publication.

  1. I have read the entire manuscript and my initial comment is that manuscript is poorly written. I have significant concerns about the grammar and vocabulary of the manuscript; therefore, improvement of the language is highly needed.

  1. The structure of the abstract should be improved, as well as the lack of several aspects that should be included in this section. Most of the abstracts contain confusing and uninformative sentences. Please give more precise objectives here (such as in the Abstract) and put the main finding of the study in the abstract.

  1. Introduction grammatical issues appear to be most prevalent in the introduction, making for very confusing reading. Further, the introduction is long but has no clear thread.

  1. Please cite the reference in line no. 37 and 38.

  1. General note: the figures in this section are quite low resolution and difficult to make out. Higher-resolution versions will be needed for publication, for example, Figure 1b, c, and Figure S1a.

  1. 132-142 lines cite the reference of the author, sometime web links may not work.
  2. 148-149 lines, how did the author select the timing for the stress treatments? No literature has been cited there.

  1. 153-155 Please describe the RT-PCR profile used by the author in this study

 the 

  1. qRT-PCR methodology provided is also very vague and confusing. Please provide more details like what the calibrator used in the study. I assume the authors have used the control as the calibrator. If so, the authors should not include the control within the bar graph as it represents the fold change between the treated vs control and a fold change of “1” for the ‘control’ doesn’t make any sense.  Also, would be good to provide details on what reagents (details of probes used, if any, if SYBR was used then details for that, etc.) .
  2. Discussion - many times references are made to the information given in the Introduction section (sometimes more general information). It would be good to discuss especially the results and critically, ie. Which can cause differences in the results of authors and other articles.

  1. Make one hypothetical figure, which depicts the findings of this study.

Author Response

Author‘s notes to reviewer are shown in attachment “Response to Reviewer 3 Comments”

Round 2

Reviewer 1 Report

The revised version of the paper by Wang et. al. has been significantly improved and might be considered for publication.

Authors are suggested to crosscheck the entire manuscript for typo errors.

Reviewer 3 Report

Dear Editor,

Thank you for providing the opportunity to review the revised manuscript. The manuscript is improved considerably after revision according to the reviewer's comment. Now this study is a suitable contribution to the gene. I recommend the manuscript for publication.

Thank you

With best regards